# Phenolic concentrations and carbon/nitrogen ratio in annual shoots of bilberry (*Vaccinium myrtillus*) after simulated herbivory

Marcel Schrijvers-Gonlag[1]*, Christina Skarpe[1], Riitta Julkunen-Tiitto[2], Antonio B. S. Poléo[1]

**1** Campus Evenstad, Faculty of Applied Ecology, Agricultural Sciences and Biotechnology, Inland Norway University of Applied Sciences, Koppang, Norway, **2** Department of Environmental and Biological Sciences, Faculty of Science, Forestry and Technology, University of Eastern Finland, Joensuu, Finland

* marcel.schrijversgonlag@inn.no

**Data Availability Statement:** The datasets and scripts used for the presented analyses are stored in the DataverseNO database and available at https://doi.org/10.18710/U8S3XX.

## Abstract

Herbivory can be reduced by the production of defense compounds (secondary metabolites), but generally defenses are costly, and growth is prioritized over defense. While defense compounds may deter herbivory, nutrients may promote it. In a field study in boreal forest in Norway, we investigated how simulated herbivory affected concentrations of phenolics (generally a defense) and the carbon/nitrogen (C/N) ratio in annual shoots of bilberry (*Vaccinium myrtillus*), a deciduous clonal dwarf shrub whose vegetative and generative parts provide forage for many boreal forest animals. We measured concentrations of total tannins, individual phenolics, nitrogen and carbon following several types and intensities of herbivory. We identified 22 phenolics: 15 flavonoids, 1 hydroquinone and 6 phenolic acids. After high levels of herbivory, the total tannin concentration and the concentration of these 22 phenolics together (called total phenolic concentration) were significantly lower in bilberry annual shoots than in the control (natural herbivory at low to intermediate levels). Low-intensive herbivory, including severe defoliation, gave no significantly different total tannin or total phenolic concentration compared with the control. Many individual phenolics followed this pattern, while phenolic acids (deterring insect herbivory) showed little response to the treatments: their concentrations were maintained after both low-intensive and severe herbivory. Contrary to our predictions, we found no significant difference in C/N ratio between treatments. Neither the Carbon:Nutrient Balance hypothesis nor the Optimal Defense hypotheses, theories predicting plant resource allocation to secondary compounds, can be used to predict changes in phenolic concentrations (including total tannin concentration) in bilberry annual shoots after herbivory: in this situation, carbon is primarily used for other functions (e.g., maintenance, growth, reproduction) than defense.

## Introduction

Bilberry (*Vaccinium myrtillus* L.) is an abundant species in boreal forests in Fennoscandia [1–3], and a key understory component influencing soil properties and forest regeneration and

**Funding:** This study is a part of the BEcoDyn project, supported by Inland Norway University of Applied Sciences and by a grant from the Norwegian Research Council to Harry P. Andreassen (who sadly passed away in 2019) (NFR project 221056; https://prosjektbanken. forskningsradet.no/en/project/FORISS/221056). MSG received a grant from the Extensus Foundation. Work done by MSG was only partly funded. The Norwegian Research Council (https:// www.forskningsradet.no/en/) and the Extensus Foundation (http://www.extensus.nu/index_eng. php) had no role in study design, data collection and analysis, decision to publish, or preparation of the manuscript.

**Competing interests:** The authors have declared that no competing interests exist.

succession [4]. This deciduous clonal dwarf shrub has evergreen shoots and grows on nutrient-poor soil by ramets, i.e., orthotropic shoots branching from buds on a rhizome [1, 2, 5–8]. In this study, we called the aboveground orthotropic shoot (the stem) including side shoots and leaves a ramet.

Vegetative and generative parts of bilberry are important forage for many mammal, bird, and insect species in all seasons [9–16]. One of many strategies to minimize herbivory on vegetative plant tissue is the production of defense compounds (defense metabolites, secondary compounds) by plants [17, 18]. Under resource limitation, trade-offs occur among growth, maintenance, storage, reproduction, and defense in plants [19, and references herein]. Some of the theories that predict plant resource allocation to secondary compounds are relevant for bilberry-herbivore interactions. In this study we focused on the Optimal Defense hypotheses (consisting of several (sub)hypotheses, see [20]) and the Carbon:Nutrient Balance hypothesis.

The Optimal Defense hypotheses predict that production of inducible defenses is low when herbivory is absent or nearly absent and increases when herbivory is present, as defenses are costly [20–28]. In general, however, growth is prioritized over defense [19, 29–32, but see 33, and references herein]. Therefore, a severe loss of photosynthetic tissue may not allow for production of defense compounds and may even lead to the breakdown of existing defenses, resulting in lower resistance to herbivory [34–37].

Like many deciduous woody species growing on nutrient-poor soils, bilberry stores carbon in roots and other woody tissue, like stems [19, 30, 38, 39]. The Carbon:Nutrient Balance (CNB) hypothesis predicts a mobilization of these carbon reserves and an increase in the level of carbon-based defense compounds in bilberry after herbivory on shoots and leaves [19, 20, 38, 40]. Phenolic compounds (phenolics) are primarily composed of carbon [41] and can act as such carbon-based defense compounds reducing herbivore performance and herbivory [42–47]. Phenolics include tannins (condensed tannins or proanthocyanidins and hydrolyzable tannins), flavonoids and other small molecular mass phenolics, including cinnamic acids [41, 42, 48]. Many different phenolics have been identified in bilberry stems, shoots, leaves, berries, and rhizomes [49–55]. We expected that the effects of tissue damage, resulting from herbivory or other causes, on phenolic concentration in bilberry vary depending on several factors: the damage type (whether leaves, shoots or the whole ramet is damaged), damage intensity, and the level the actual phenolic can function as a defense against herbivores, as different phenolics have multiple biological functions and efficacy [56–63].

Defense compounds may deter herbivory, while nutrients may promote it [18, 64–68]. Nitrogen concentration in bilberry, which is often used as a proxy for nutrient concentration, increases after browsing in several woody species, often regardless of soil productivity [69–73]. Pruning, the partial or complete removal of stem and/or shoots, reduces bud numbers and increases the root:shoot ratio. This results in decreased competition for nutrients among meristems, causing an increase in new plant tissue nutrient concentration [32, 74–78]. On the other hand, severe defoliation results in a loss of nitrogen [79], or at least in the loss of proportionally more nutrients than carbon, as most nutrients are found within the foliage of deciduous species in the growing season [19]. Furthermore, severe defoliation results in increased fine root mortality [80, 81]. This leads to reduced nutrient absorption which results in a decreased nutrient concentration, especially on nutrient-poor soils [82, 83]. Therefore, we expected that the effects of tissue damage, due to herbivory or other causes, on nitrogen concentration in bilberry vary, depending on type and intensity of damage.

Most research on phenolics in bilberry has focused on berries, although studies on leaves, shoots and stems have been conducted [49, 50, 54, 84–86]. Previous studies of herbivory, nutritional quality and defense responses of bilberry shoots and leaves did not involve controlled clipping experiments, nor measurements of phenolic, nitrogen and carbon concentrations in

annual shoots [87–92]. After herbivory, we expected a measurable response in the young tissue of annual shoots [21, 39, 93]. For these reasons, we investigated how simulated herbivory affected phenolic concentrations and the carbon/nitrogen (C/N) ratio, often used as indicator of plant nutritional quality [94, 95], in bilberry annual shoots. We measured total tannin concentration and concentrations of individual phenolics, nitrogen and carbon in bilberry annual shoots after several types and intensities of simulated herbivory. Persson and colleagues [55] performed a simulated browsing study on bilberry investigating responses in phenolic and nitrogen concentrations and C/N ratio in leaves and leafless shoots. Different from Persson and colleagues, who performed different levels of simulated moose (*Alces alces* L.) browsing only, we used three types of simulated herbivory, representing herbivory by large ungulates (eating ramets), herbivory by smaller mammals, birds and insects (eating annual shoots) and herbivory by insects (eating leaves). Our study was performed under ambient herbivory conditions in boreal forest in southeastern Norway in 2014.

Our objective was to examine how different herbivory types (ramet herbivory, annual shoot herbivory, leaf herbivory) and intensities affect the concentration of phenolics (total tannins as well as several small molecular mass phenolics) and nitrogen (nutritional quality) in bilberry annual shoots. We compared our simulated herbivory (from here often just called herbivory) with ambient herbivory, which was at a low to intermediate level. We considered our results in the context of the plant defense theories described above. We predicted that in bilberry annual shoots, the concentration of:

I. phenolics is, at low to intermediate herbivory levels, positively correlated with intensity of herbivory;

II. phenolics is, at high herbivory levels, lower than without herbivory;

III. nitrogen is, at low to intermediate herbivory levels, positively correlated with intensity of herbivory, i.e., the C/N ratio is negatively correlated with intensity of herbivory;

IV. nitrogen is, at high herbivory levels, lower, i.e., the C/N ratio is higher, than without herbivory.

## Methods

### Study area

We conducted our study in coniferous boreal forest at six locations (400–670 m a.s.l.) in the Østerdalen valley close to Evenstad (latitude 61.43 ˚N, longitude 11.08 ˚E) in southeastern Norway in 2014. In this year, mean annual temperature was 4.8 ˚C and total precipitation was 896 mm [96]. The forest was owned by the Norwegian state-owned land and forest enterprise Statskog SF (www.statskog.no), who granted permission to do the experiment, including sampling bilberry plants.

### Study design

**Field treatments.** At each location, we used four lines, more or less parallel and spaced by ten m, to select bilberry ramets with approximately two m distance between consecutive ramets (Fig 1). Along each line, we selected 33 or 34 ramets at the beginning of the growing season (May) and marked them with steel wire. Selected ramets had at least ten shoots longer than 1.0 cm from the previous growing season (annual shoots from 2013, S1 File), and no visual signs of extensive herbivory (most ramets had some past herbivory signs), so the initial herbivory level for all ramets was low. In total we selected 135 ramets at each location. We

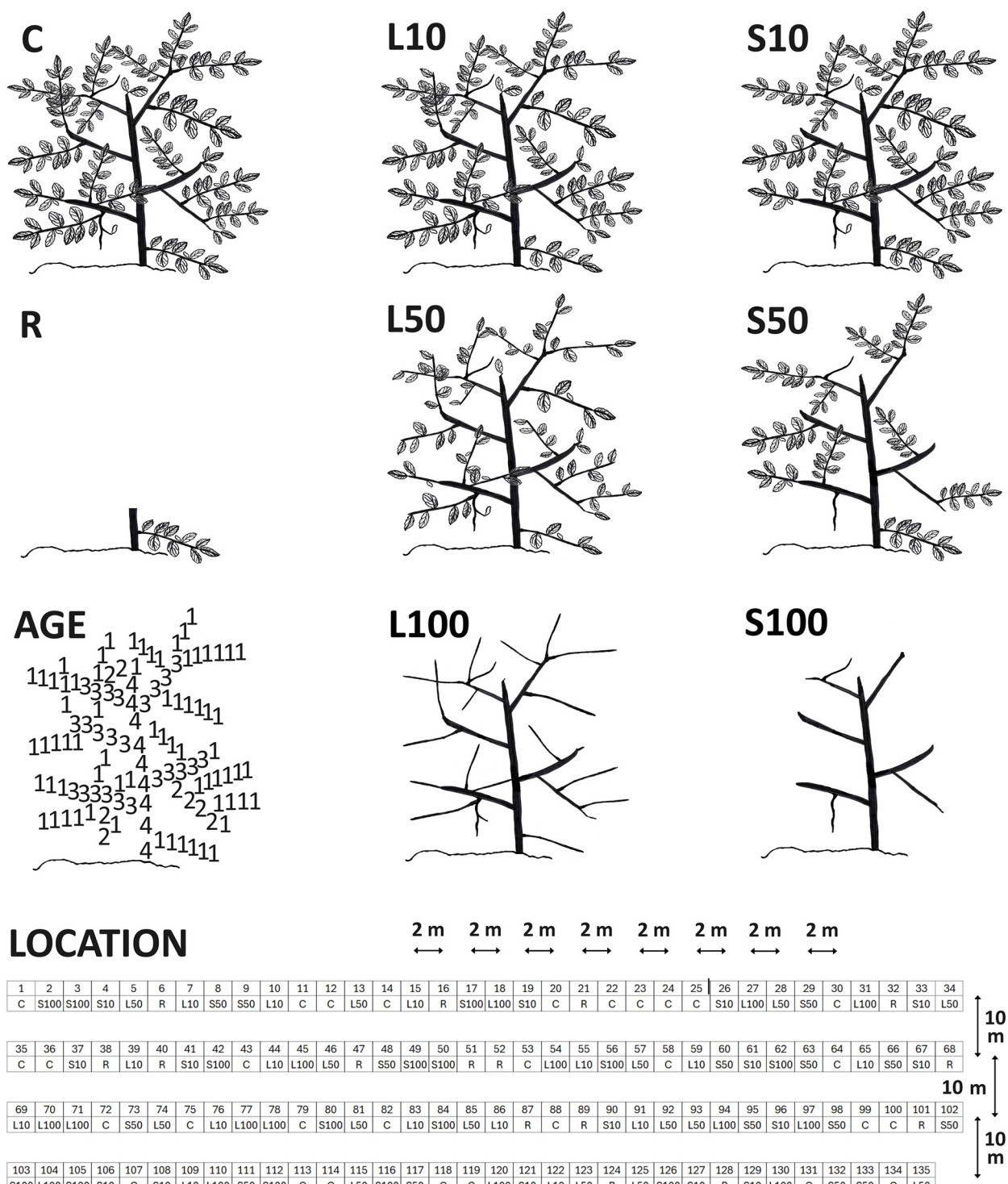

**Fig 1. Study design.** Schematic bilberry ramet, eight treatments (see text): C (control), three leaf treatments (L), three shoot treatments (S), one ramet treatment (R). 21 annual shoots are visible (1 at the top of the stem, 10 at each side of the stem). AGE: the stem and all shoots (same schematic bilberry as in the treatments) are depicted with numbers: the stem is at least four years old and indicated with 4, shoots at least three years old are indicated with 3, shoots at least two years old are indicated with 2, annual shoots (with leaves) are indicated with 1. NOTE: the upper three-year-old shoot at the right side of the stem and the middle three-year-old shoot at the left side of the stem may also be two years old. LOCATION: at four lines, 34 (upper three lines) and 33 (lower line) ramets are selected: 30 control ramets and 15 ramets for every one of the other 7 treatments, randomly appointed. Approximately 2 m between each ramet and 10 m between each line. This location is one of six locations (namely: "Imsdalen 1"). Drawing: Marcel Schrijvers-Gonlag.

divided the ramets within each location randomly (S1 File) into four treatment groups: control (n = 30), abbreviated to C, representing ambient, initially low, herbivory only; 'leaves cut' (n = 45), abbreviated to L, representing additional herbivory by insects; 'annual shoots cut' (n = 45), abbreviated to S, representing additional herbivory by insects and small-sized vertebrates; and 'ramet cut' (n = 15), abbreviated to R, representing additional herbivory by large ungulates. At all six locations, we removed leaves by hand (treatment L) at three different intensities: 10% from 15 ramets, 50% from another 15 ramets and 100% from the remaining 15 ramets (S1 File). At five locations, we removed annual shoots by hand (treatment S) at similar intensities (10%, 50% and 100%; n = 15 for each), and we cut the ramet in treatment R by removing 90% of the ramet with garden scissors (Fig 1). In total this resulted in eight treatments: 'control' (C: ambient, initially low, herbivory), 'leaves cut' (3 intensities: L10, L50, L100), 'annual shoots cut' (3 intensities: S10, S50, S100), 'ramet cut' (R). Ramets in C that experienced severe herbivory between selecting and harvesting, were excluded from our analyses: therefore, all control ramets experienced herbivory at low to intermediate levels (ambient herbivory). The shoots were removed and ramets cut on 24–27 May and leaves were removed in the period 21 June– 2 July. In our experiment, we considered C as herbivory at the lowest level. Within L, L10 represented leaf herbivory at a low level, L50 represented leaf herbivory at an intermediate level and L100 represented leaf herbivory at a high level. Similar with S: within S, S10 represented annual shoot herbivory at a low level, S50 represented annual shoot herbivory at an intermediate level and S100 represented annual shoot herbivory at a high level. We considered R as herbivory at the highest level and S100 as herbivory at the second highest level in our experiment: judging after proportion of biomass removed, these two treatments were the two most severe herbivory treatments in our study.

**Bilberry sampling.** All ramets (n = 750) were harvested towards the end of the growing season (19–28 August) by cutting the stem at ground level with garden scissors. The ramets were dried for minimum 48 hours in a drying oven (Binder FED 720 E2, Germany) at 30 ˚C, before the dried ramets were stored in a dark and dry place at room temperature. From each location, we randomly selected a minimum of five dried ramets from each herbivory treatment (including control), resulting in 232 ramets in total. From each of these ramets, we randomly selected five annual shoots (S1 File), continued drying these annual shoots for minimum 24 hours at 30 ˚C and stored them in a dark and dry place at room temperature, prior to preparation and analyses of tannin, phenolic, carbon and nitrogen concentrations.

## Chemical analyses

**Bilberry shoots.** Before analyses of acetone-soluble tannins, methanol-soluble phenolics, and total carbon and nitrogen, the shoots were cut in fragments of maximum 0.5 cm and for each ramet we transferred these subsamples to a 2 ml or 7 ml vial with three stainless steel beads (2.8 mm) to pulverize the tissue; with large subsamples (approximately 200, 300 and 400 mg; all weight measurements in this study: scale Sartorius SE2, d = 0.1 μg) we used four, five or six beads, respectively. The shoot fragments were pulverized by the beads using a Precellys 24 homogenizer (Bertin Technologies, France): 25 s at 5500 rpm, 15 times with two minutes in between. When handling the shoots, we used disposable latex gloves.

**Shoot tannins.** The shoots were analyzed for acetone (70%)-soluble tannins (e.g., hydrolyzable tannins and polymeric condensed tannins (proanthocyanidins)) [97, 98: S1, 99] with a spectrophotometer (Spectronic 20 Genesys; Spectronic Instruments, USA). We slightly adjusted the acid butanol assay for proanthocyanidins [100] to measure tannins in our subsamples (S2 File). To relate tannin concentration in our subsamples to measured absorbance

(at 550 nm) we built a standard reference curve, using Sephadex LH-20 (GE Healthcare Bio-Sciences AB, Sweden) for tannin purification [98: S1, 101].

**Shoot phenolics.** Methanol-soluble phenolics (e.g., flavonoids and phenolic acids) were extracted from the shoots and quantified using high performance liquid chromatography (HPLC) with injection volume 10 μl (Agilent series 1100) and identified using a UHPLC quadrupole time-of-flight liquid chromatograph–mass spectrometer (Agilent Technologies, 6540 UHD Accurate-Mass Q-TOF LC/MS, 1290 Infinity) as described by Nissinen and colleagues [102] (S3 File). Compounds that could not be identified were not used in this study. We used D(-)-Salicin min. 99% CHR (Aldrich-Chemie, West-Germany) in methanol (100%) as an internal standard in two out of five subsamples to evaluate extraction efficiency (S3 File).

**Shoot carbon and nitrogen.** The shoots were analyzed for carbon and nitrogen (total concentration (mg/g, dry weight) after destruction; micro CN-analyzer (thermo), Chemical Biological Soil Laboratory (quality system based on the ISO-17025 standard), Wageningen University, July 2016).

## Statistical analyses

The total tannin absorbance measurements were averaged per subsample and with the standard reference curve and subsample weight these subsample means were converted to concentrations (mg tannins/g shoots, dry weight), which were used in further analyses. In our HPLC analyses we used the concentration (mg/g, dry weight) of every identified phenolic as the response variable in our modeling, calculated as: ((rf x area) / weight) / (inj / tot) where rf is the HPLC response factor for the actual phenolic at the used wavelength, area is the peak area in the HPLC result table (mAU*s) at the used wavelength, weight is the initial shoot subsample weight (mg), inj is the HPLC injection volume (10 or 15 μl) and tot is the total volume (600 μl) in which the subsample was dissolved (300 μl methanol + 300 μl purified water, S3 File). The HPLC response factor is the ratio between the concentration of a specific compound (mg/g) and the response of the detector (area: mAU*s) to this compound at a specific wavelength; we used response factors previously determined using standards with known concentrations (S1 Table). Before analyses, phenolic concentrations were converted to 100% to recover losses in the extraction procedure (S3 File). When no value in the HPLC result table was present for a phenolic, we used a concentration of 0 mg/g, although often a small peak on the HPLC chromatogram was visible.

Differences between treatments were investigated with a one-way ANOVA test. In all ANOVA analyses we used equal sample sizes across groups, to avoid inflation of error rates and to guarantee homogeneity of variance [103]. If necessary, samples were randomly removed to obtain balanced sample sizes. We used the total tannin concentration, the concentration of each identified phenolic and the concentration of all identified phenolics together as response variables in predictions I and II. We also used a one-way ANOVA test to investigate differences between treatments on the response variable C/N ratio (predictions III and IV). When the ANOVA test indicated a significant difference (we used a significance level of 5%), differences between groups were investigated with Tukey's HSD post-hoc test. We used the package 'emmeans' in the software 'R' to calculate some general statistics and to further investigate the relationship between several response variables and treatments [104, 105]. Figs 2–5 were created with the R-package 'svglite' [106] and the software 'Inkscape' (version 1.2.1). All model analyses were performed in R (version 4.1.2, 4.2.2, 4.2.3 and 4.3.1) [105].

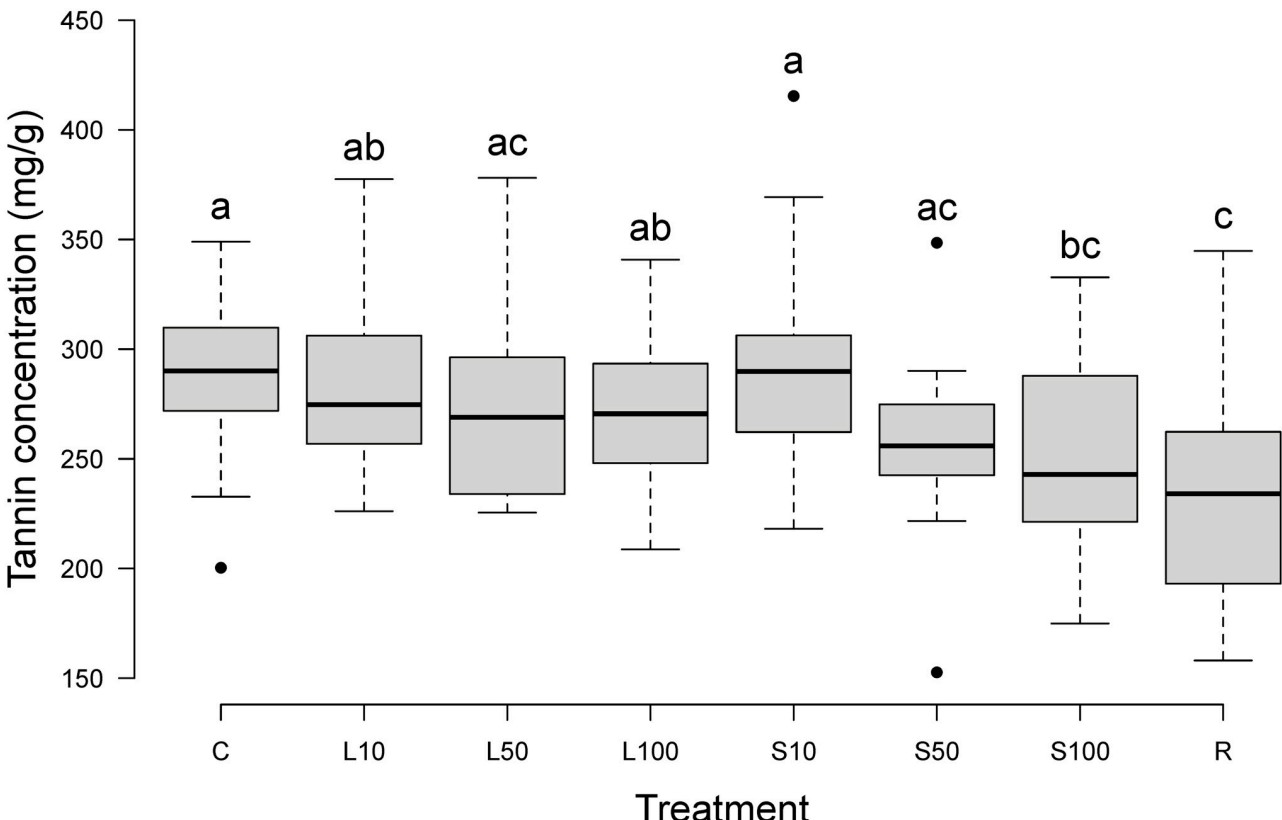

**Fig 2. Tannin concentration in bilberry annual shoots after simulated herbivory.** Boxplot with total concentration (mg/g, dry weight), n = 200, every treatment n = 25. Treatments: see text. The bottom and top of each box indicate the first and third quartiles. Bold horizontal lines within each box indicate median values. The plot whiskers extend to the most extreme data point which is no more than 1.5 times the interquartile range away from the box; extreme data points more than 1.5 times the interquartile range away from the box are indicated with black points. Treatments with the same letter above the box are not different from each other ($P > 0.05$).

## Results

### Shoot tannins

On average, bilberry annual shoots consisted of 25–30% tannins (dry weight; Table 1). The intensity of herbivory affected tannin concentrations (ANOVA: $F_{7,192} = 6.18$, $P < 0.001$; Fig 2; Table 1). S100 and R resulted in significantly lower tannin concentrations than C. All other treatments did not differ significantly from each other nor from C (Fig 2).

### Shoot phenolics

We identified 22 phenolics: 15 flavonoids, 1 hydroquinone and 6 phenolic acids (Fig 3). The recovery of the internal standard was around 95%: min = 58.9%, mean = 94.5%, max = 107.5%, sd = 6.1% (n = 90). In one subsample the recovery of the internal standard was 58.9%, quite different from all others. Therefore, we excluded this subsample from the phenolic analyses. Without this subsample, the recovery of the internal standard changed to: min = 83.1%, mean = 94.9%, max = 107.5%, sd = 4.7% (n = 89).

The phenolic concentration of all these 22 identified phenolics together (analyzed together) is from here called <u>total phenolic</u> concentration. On average, almost 10% of bilberry annual shoots consisted of these 22 phenolics (dry weight; Table 1). Compared to C and L10, which

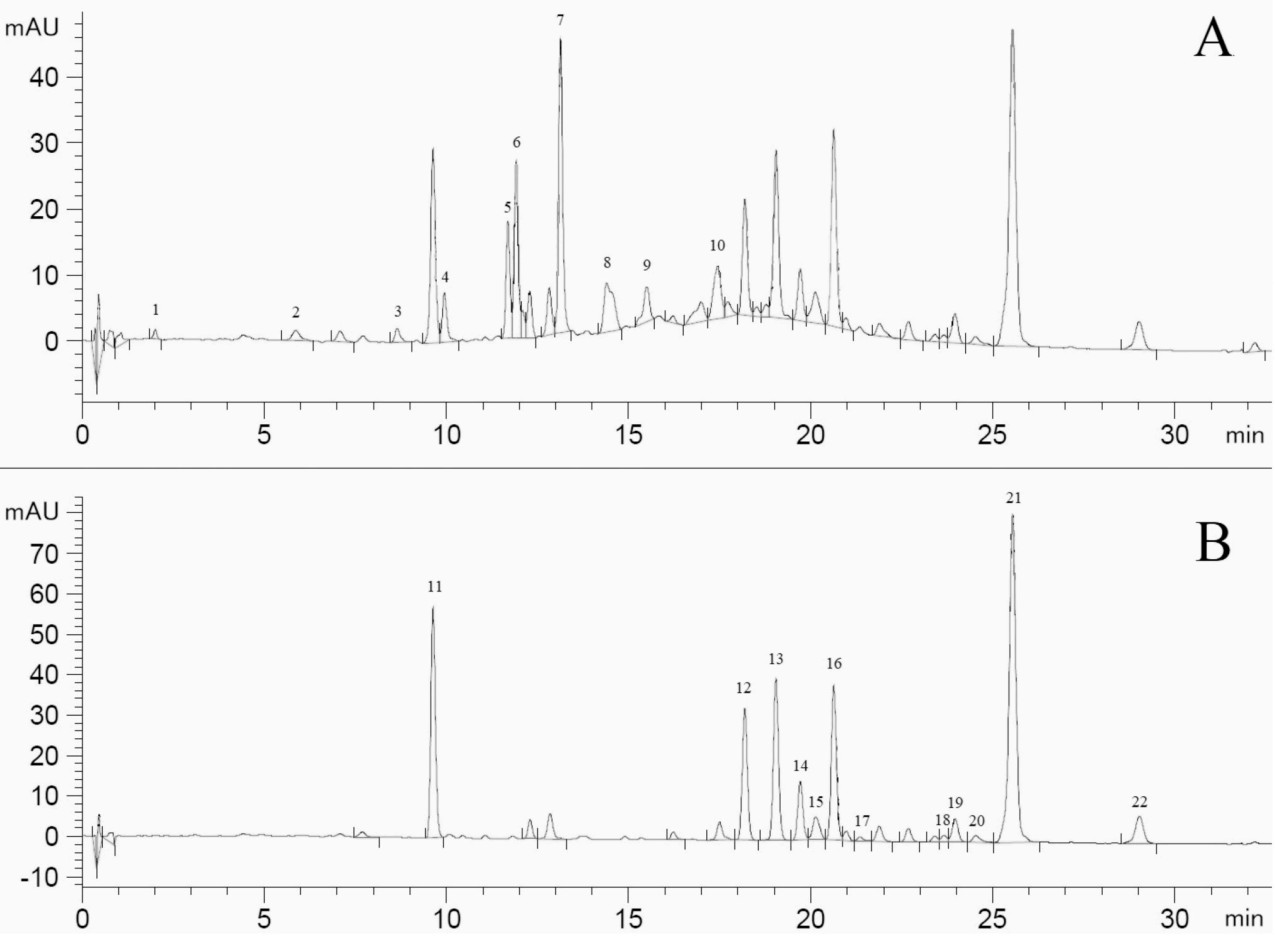

**Fig 3. HPLC chromatogram of phenolics in bilberry annual shoots.** The chromatograms shown here (**A** wavelength 280 nm, **B** wavelength 320 nm; retention time (x-axis) in minutes (min), response (y-axis) in mAU (AU = absorption units)) are from the subsample which was used to identify the peaks with mass spectrometry. Phenolics identified (for footnotes, see S1 Table): 1. protocatechuic acid derivative, 2. arbutin derivative[7], 3. gallocatechin derivative, 4. procyanidin 1, 5. procyanidin 2, 6. epicatechin (formerly called: (-)-epicatechin), 7. procyanidin 3, 8. procyanidin 4, 9. procyanidin 5, 10. procyanidin 6, 11. chlorogenic acid, 12. para-hydroxycinnamic acid derivative 1, 13. cinnamic acid derivative, 14. para-hydroxycinnamic acid derivative 2, 15. hyperin[1], 16. quercetin 3-glucuronide[5], 17. quercetin 3-arabinoside[4], 18. kaempferol 3-glucoside[2], 19. quercitrin[6], 20. isorhamnetin 3-glucoside, 21. para-hydroxycinnamic acid derivative 3, 22. monocoumaroyl-isoquercitrin[3,8].

had very similar total phenolic concentrations, all other treatments except S10 resulted in lower mean total phenolic concentrations (Table 1; S2 Table). The differences between C, L10 and S10 were not significant (Fig 4), but a significant difference in total phenolic concentration between one or more other treatments was present (ANOVA: $F_{7,192} = 6.64$, $P < 0.001$). Within L, the total phenolic concentration did not differ significantly, but it did within S (Fig 4). S100 and R resulted in significantly lower total phenolic concentrations than C. R resulted in the lowest mean (S2 Table) and median (Fig 4) total phenolic concentration.

As the total phenolic concentration is the sum of all identified phenolic concentrations, many of these <u>individual phenolics</u> showed a similar pattern: R resulted in the lowest mean phenolic concentration in 15 phenolics (68%). Considering R and S100 together, this number increased to 19 phenolics (86%) (S2 Table). Investigating significant differences between treatments, one or more herbivory treatments resulted in significantly different phenolic concentrations in nine phenolics; phenolic acids showed little response to the treatments (Table 2).

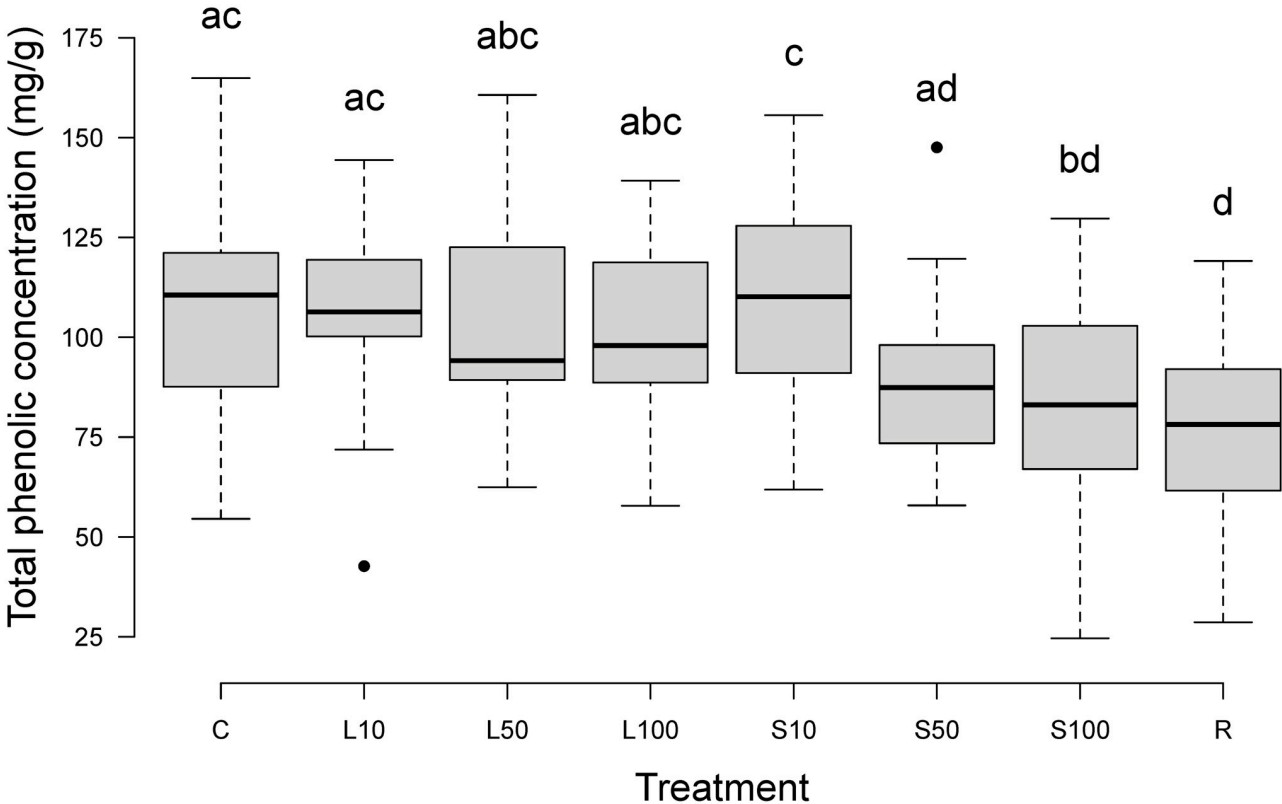

**Fig 4. Total phenolic concentration in bilberry annual shoots after simulated herbivory.** Boxplot with total phenolic concentration (mg/g, dry weight), n = 200, every treatment n = 25. Treatments: see text. The bottom and top of each box indicate the first and third quartiles. Bold horizontal lines within each box indicate median values. The plot whiskers extend to the most extreme data point which is no more than 1.5 times the interquartile range away from the box; extreme data points more than 1.5 times the interquartile range away from the box are indicated with black points. Treatments with the same letter above the box are not different from each other ($P > 0.05$).

Considering the leaf (L) removal treatments only, there were no significant differences in individual phenolic concentration between treatments, nor were any L treatments significantly different from C (Fig 5). Considering the annual shoot (S) removal treatments, in two phenolics (procyanidin 2 and 4) the phenolic concentration was significantly lower in S100 than in S10. Compared to the other treatments, S10 resulted in the highest mean phenolic concentration in eight phenolics (36%) and in the total phenolic concentration (S2 Table), although no significant difference between S10 and C was present in any of the phenolics. In three phenolics (epicatechin, procyanidin 2 and quercetin 3-glucuronide) S100 resulted in a significantly lower concentration than C (Fig 5). In seven phenolics, R resulted in a significantly lower phenolic concentration than C (Fig 5).

## Shoot carbon and nitrogen

On average, half of the bilberry annual shoots (dry weight) consisted of carbon and slightly over 1% consisted of nitrogen (Table 1). There was little variation in carbon and nitrogen concentration and in C/N ratio between treatments (Table 1). We found no significant difference in C/N ratio between treatments (ANOVA: $F_{7,192} = 0.40$, $P = 0.90$). Analyses of carbon concentration and nitrogen concentration yielded similar results: no significant difference between

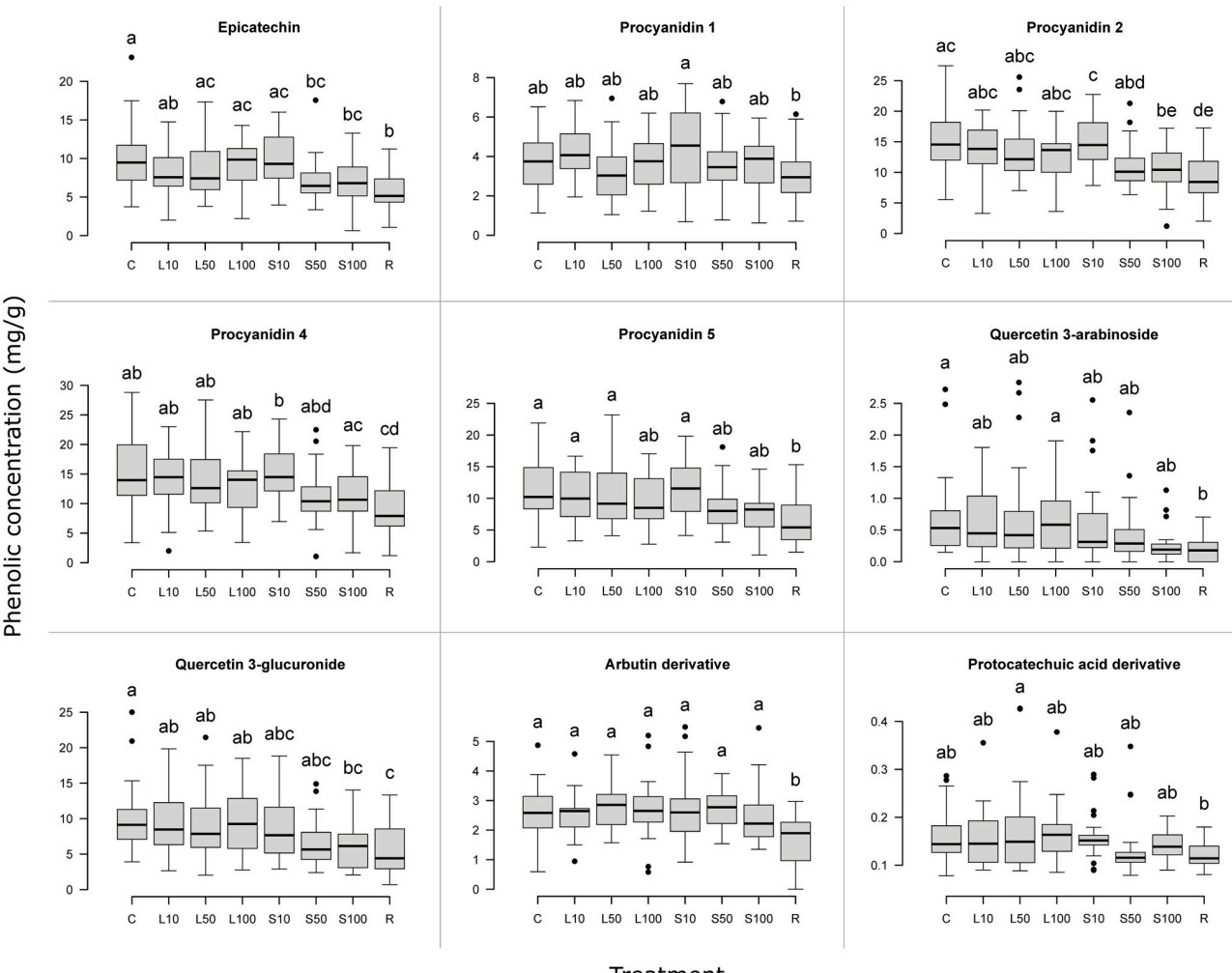

**Fig 5. Individual phenolic concentration in bilberry annual shoots after simulated herbivory.** Boxplots with individual phenolic concentration (mg/g, dry weight). Only phenolics with at least one treatment significantly different from another treatment are shown (n = 200, every treatment n = 25). Treatments: see text. The bottom and top of each box indicate the first and third quartiles. Bold horizontal lines within each box indicate median values. The plot whiskers extend to the most extreme data point which is no more than 1.5 times the interquartile range away from the box; extreme data points more than 1.5 times the interquartile range away from the box are indicated with black points. Treatments with the same letter above the box are not different from each other (*P* > 0.05).

treatments (carbon: ANOVA: $F_{7,192}$ = 1.04, *P* = 0.40; nitrogen: ANOVA: $F_{7,192}$ = 0.37, *P* = 0.92).

## Discussion

### Shoot tannins and total phenolics

The two most severe herbivory treatments, concerning removed biomass, resulted in significantly lower tannin and total phenolic concentrations than the control, while less severe herbivory treatments did not differ significantly from the control. These results support our second prediction but suggest that little to intermediate loss of photosynthetic tissue does not increase carbon-based defense compound concentrations in bilberry, contrary to our first prediction. Several factors may contribute to these results.

**Table 1. Tannins, phenolics, carbon, nitrogen, and C/N in bilberry annual shoots after simulated herbivory.** Mean concentration (mg/g ± se (standard error of the sample mean), dry weight) of tannins, 22 phenolics (see text), carbon (C) and nitrogen (N) and mean C/N ratio (± standard error), per treatment (see text). For all treatments together (All) also the sd (standard deviation of the sample) is given. Number of observations between parentheses.

| Treatment | Tannins | Phenolics | Carbon (C) | Nitrogen (N) | C/N |
|---|---|---|---|---|---|
| C | 291.1 ± 7.1 (33) | 106.4 ± 4.1 (33) | 498 ± 2.0 (33) | 12.27 ± 0.36 (33) | 41.9 ± 1.2 (33) |
| L10 | 278.2 ± 7.1 (33) | 106.1 ± 4.2 (32) | 497 ± 2.0 (33) | 12.40 ± 0.36 (33) | 40.9 ± 1.2 (33) |
| L50 | 275.7 ± 7.2 (32) | 100 9 ± 4.2 (32) | 498 ± 2.1 (32) | 12.64 ± 0.37 (32) | 40.1 ± 1.2 (32) |
| L100 | 274.4 ± 7.2 (32) | 101.4 ± 4.2 (32) | 498 ± 2.0 (32) | 12.26 ± 0.37 (32) | 41.7 ± 1.2 (32) |
| S10 | 292.8 ± 8.2 (25) | 110.2 ± 4.7 (25) | 500 ± 2.4 (25) | 12.17 ± 0.42 (25) | 42.1 ± 1.4 (25) |
| S50 | 257.1 ± 8.2 (25) | 88.6 ± 4.7 (25) | 498 ± 2.4 (25) | 12.94 ± 0.42 (25) | 39.6 ± 1.4 (25) |
| S100 | 249.3 ± 8.0 (26) | 84.9 ± 4.6 (26) | 504 ± 2.3 (26) | 12.76 ± 0.41 (26) | 40.7 ± 1.4 (26) |
| R | 235.7 ± 8.0 (26) | 75.1 ± 4.6 (26) | 500 ± 2.3 (26) | 12.62 ± 0.41 (26) | 41.3 ± 1.4 (26) |
| All | 270.5 ± 2.9, sd = 44.1 (232) | 97.5 ± 1.7, sd = 25.9 (231) | 499 ± 0.8, sd = 11.8 (232) | 12.50 ± 0.14, sd = 2.08 (232) | 41.0 ± 0.5, sd = 6.9 (232) |

**Table 2. Differences in individual phenolic concentration in bilberry annual shoots between simulated herbivory treatments.** The number of differences (#) between treatments ($P < 0.05$) is given for 22 phenolics separately, and for all 22 phenolics analyzed together (n = 200, every treatment n = 25).

| Flavonoids | # |
|---|---|
| Epicatechin | 6 |
| Gallocatechin derivative | ns |
| Hyperin | 0 |
| Isorhamnetin 3-glucoside | ns |
| Kaempferol 3-glucoside | ns |
| Monocoumaroyl-isoquercitrin[1] | ns |
| Procyanidin 1 | 1 |
| Procyanidin 2 | 8 |
| Procyanidin 3 | ns |
| Procyanidin 4 | 6 |
| Procyanidin 5 | 4 |
| Procyanidin 6 | ns |
| Quercetin 3-arabinoside | 2 |
| Quercetin 3-glucuronide | 5 |
| Quercitrin | ns |
| *Sum flavonoids* | *32* |
| **Hydroquinones** | |
| Arbutin derivative | 7 |
| *Sum hydroquinones* | *7* |
| **Phenolic acids** | |
| Chlorogenic acid | ns |
| Cinnamic acid derivative | ns |
| Para-hydroxycinnamic acid derivative 1 | ns |
| Para-hydroxycinnamic acid derivative 2 | ns |
| Para-hydroxycinnamic acid derivative 3 | ns |
| Protocatechuic acid derivative | 1 |
| *Sum phenolic acids* | *1* |
| *Sum all individual phenolics* | *40* |
| **All 22 phenolics together** | 9 |

[1]Monocoumaroyl-isoquercitrin: identification uncertain.

If the ANOVA test result $P < 0.05$ but the Tukey's HSD test gave only $P_{adj}$ values $> 0.05$ a '0' is shown in the table.

ANOVA test results $P > 0.05$ show 'ns' in the table (irrespective of the Tukey's HSD test result).

First, not all phenolics respond to herbivory as predicted by the CNB hypothesis, as many phenolics compete with proteins for the nitrogen containing precursor phenylalanine as described by Jones & Hartley [107] in their Protein Competition Model (PCM) [42, 108, 109]. As the boreal forest is a nitrogen-limited ecosystem [110–113], competition for nitrogen between biosynthesis of proteins and of many phenolics is expected in boreal forest ecosystems. Therefore, the nutrient-poor soil may not provide sufficient nitrogen for bilberry to increase these secondary compound concentrations while continuing protein demanding primary processes as growth and reproduction.

As bilberry is a clonal plant, connected ramets may translocate compounds from nondefoliated to defoliated ramets, as has been documented in perennial graminoid species and herbs [114–116, and references herein]. Translocation of carbohydrates or even phenolics between connected ramets may be another reason for the absence of a correlation between leaf herbivory and subsequent phenolic concentrations in bilberry annual shoots. Also, high fine root mortality may not, or to a lesser extent, occur in clonal bilberry, which possibly translocates carbohydrates between connected ramets to maintain its root activity.

Furthermore, our simulated leaf herbivory (mechanical wounding by hand) is not equivalent to leaf herbivory by insects, birds, or small mammals [27, 117–125, reviewed by 126]. Although most of these studies indicate that simulated leaf and shoot herbivory performed by mechanical wounding induces a less pronounced plant response, such simulated herbivory generally does induce a plant response [see the aformentioned references and 87, 127, 128], as the general response of plants to wounding and herbivore damage is essentially the same [129]. This is particularly true in nutrient-poor sites [130], like our study system. Nevertheless, this indicates that bilberry responses to simulated leaf herbivory may differ from responses to natural herbivory, which can further contribute to our observed absence of a correlation between simulated leaf herbivory and subsequent phenolic concentrations in bilberry annual shoots.

Additionally, this observed absence can be due to other reasons. A response can have been counteracted by transport of existing phenolics from shoots to leaves, as some plant species store phenolics in shoots which are transported to leaves following herbivory [127]–although such reallocation of phenolics may not be very important [107]. Furthermore, the time between our leaf herbivory treatments and bilberry ramet harvesting was 48–68 days. Possibly, bilberry only responds with a short-term response that was no longer detectable after 48 days. For instance, in another woody species, the condensed tannin concentration returned to pre-herbivory values less than 66 hours after herbivory [131, see also 132]. An alternative possible reason is a very delayed response: responses remain undetectable until at least 68 days after the treatment. This last option seems very unlikely in terms of plant fitness, but cannot be ruled out with the data available. Experiments measuring how long induction lasts in different bilberry tissues are needed to support or reject these speculations.

Another possible reason for our observed results is that an herbivory-induced change in phenolics occurs in other plant parts, e.g., leaves, and is not detectable in annual shoots. This seems unlikely, as Persson and colleagues found that bilberry leaves and bilberry leafless shoots were comparable in their response to simulated moose herbivory, at least for flavonoids and condensed tannins [55]. In contrast with our results, Persson and colleagues found an increase in flavonoid and condensed tannin concentration in bilberry shoots with increasing simulated moose herbivory. Possibly their results were influenced by a side-effect of the treatment: a more open canopy resulted in more solar radiation which could have induced production of secondary compounds, as has been found and discussed in other studies [50, 55, 133–139, and references herein].

## Shoot individual phenolics

In seven phenolics, R resulted in a significantly lower phenolic concentration than C, while in fifteen phenolics no significant difference between the control and other treatments was present. The concentrations of all phenolic acids found in our study were unaffected by severe herbivory. Interestingly, these phenolics are known to deter herbivory by insects: all reduce larval growth rate, some also promote larval mortality and chlorogenic acid even shows strong antinutritive properties against various invertebrate herbivores, including adult beetles and grasshopper nymphs [45, 140–147]. This indicates that bilberry responds to severe herbivory by maintaining concentrations of phenolics which deter herbivory on a certain level. As we could not find information about biological functions related to herbivory for other specific phenolics identified in our study, we don't know how bilberry responds to severe herbivory in the case of phenolics which promote herbivory (possibly by decreasing their concentrations?). Herbivory experiments with specific phenolics are necessary to support or reject these speculations.

We did not find (+)-catechin in our bilberry annual shoots, as has been found in other bilberry studies [49, 50]. In the HPLC chromatogram (Fig 3), (+)-catechin, if present, comes shortly after chlorogenic acid. This means that when a large quantity of chlorogenic acid is present, as with our subsamples (Fig 3, S2 Table), the chlorogenic acid peak overlaps with the peak of (+)-catechin and it is not possible to separate the latter from the former, especially when only little (+)-catechin is present. Therefore, unidentified amounts of (+)-catechin may have been present in our subsamples, but if so, (+)-catechin was present in much lower amounts than epicatechin (S2 Table).

## Shoot carbon and nitrogen

The carbon and nitrogen concentrations and C/N ratio in our study are comparable with results from other studies [6, 50, 136, 148, 149] but differ from bilberry nitrogen concentrations found by Selås and colleagues [150]. Our results show that both the carbon and nitrogen concentration, as well as the C/N ratio, in bilberry annual shoots are not affected by herbivory. These findings do not support our predictions III and IV. Apparently, mechanisms that either increase or decrease nutrient concentration after herbivory (see Introduction), cause this overall result. Additionally, in clonal bilberry carbohydrates may be translocated from source ramets to connecting ramets under herbivory pressure, and to their root system, to compensate for a lack of carbon (see before). This may prevent an increase in fine root mortality and, consequently, a decrease in nutrient concentration. Thus, clonality can further explain the lack of support for our predictions III and IV.

Another possible reason is, as with phenolics (see before), that a change in C/N ratio does not occur in bilberry annual shoots but in other plant parts, e.g., leaves, as shown in other woody species [70, 72, 75, 95, 151] (although Laine and Henttonen [148] did not find a correlation between microtine density and nitrogen concentration in bilberry leaves). As we do not have data about carbon and nitrogen concentrations in plant parts other than annual shoots, we cannot rule out this possibility.

Finally, Flower-Ellis [6] reported much variation in nitrogen concentration between long, vegetative shoots and short, predominantly flowering shoots, as well as in ramets from different ages and positions in the stand (causing variation in light and water conditions). Such variation may obscure effects from herbivory.

## Study design

In this study, we removed annual shoots in four treatments, at different intensities: S10, S50, S100 and R. Only with the last two treatments (S100 and R), all (or almost all in some R

treatments) annual shoots were removed. Approximately three months after removal, we harvested annual shoots from the ramets: therefore, only with S100 and R the harvested annual shoots were all (or almost all in some R treatments) new shoots, grown after the clipping event had occurred. In all other treatments, most likely the analyzed annual shoots had all (L treatments) or partly (S10: around 90 percent, S50: around 50 percent) been present at the ramet before the clipping event took place. Interestingly, only at high herbivory levels (S100 and R), we found a significant difference in tannin concentration and total phenolic concentration compared to the control. This means that all treatments from which we analyzed many older annual shoots (from before the clipping event) did not yield a significant difference in phenolic concentration in the annual shoots compared to the control. Although many of these annual shoots probably were not fully grown at the time of clipping and therefore also their tissue had (partly) developed after the clipping event took place, this means that we analyzed annual shoots in S10, S50 and all L treatments, that were present before the clipping event occurred–at least an important part of them. If a chemical response to the treatment does not occur in older shoots but only, or mainly, occurs in new tissue (this we don't know) this shortcoming in our study design has serious consequences for our results regarding to the S10, S50 and all L treatments.

## Defense and other metabolic processes

As our results do not support our first prediction (I: phenolic concentration is, at low to intermediate herbivory levels, positively correlated with intensity of herbivory) but do support our second prediction (II: phenolic concentration is, at high herbivory levels, lower than without herbivory), we conclude that after herbivory, bilberry uses carbon primarily for functions other that defense. This is no more than a speculation, as we have no metric of growth (as total biomass or compensatory growth), or metabolic processes other than phenolic concentrations. Possibly, little herbivory may be almost inconsequential for plant fitness and responses may be absent, or non-detectable, or only morphological, not chemical. Severe herbivory may force bilberry to divert resources from other pools, as existing defense chemical compounds, to compensate for biomass losses. Experiments which specifically focus on morphological responses (as compensatory growth) and reproduction, preferably also chemical responses, after herbivory, are needed to support or reject our speculation.

## Conclusions

We conclude that neither the Carbon:Nutrient Balance hypothesis nor the Optimal Defense hypotheses can be used to predict changes in phenolic concentrations (including total tannin concentration) after herbivory in bilberry annual shoots. After herbivory, bilberry uses carbon primarily for functions other than defense (e.g., maintenance, growth, reproduction). Herbivory experiments focusing on morphological responses and reproduction are necessary to further investigate this response. Furthermore, we conclude that bilberry responds to severe herbivory by maintaining concentrations of specific phenolics, which deter herbivory, on a certain level, while decreasing concentrations of other phenolics. Herbivory experiments with specific phenolics, to clearify their function as anti-herbivore compound (i.e., do they affect bilberry's palatability to herbivores), are necessary to further investigate this response.

## Supporting information

**S1 File. Annual shoots and random selection.**
(PDF)

**S2 File. Standard reference curve and tannin color test.**
(PDF)

**S3 File. Quantifying phenolics using HPLC.**
(PDF)

**S1 Table. Response factors.**
(PDF)

**S2 Table. Phenolic concentrations after simulated herbivory.**
(PDF)

## Acknowledgments

We thank Anne Mehlhoop, Bernardo Toledo González and Marieke Gonlag-Schrijvers for lab work and analytical insights, Sinikka Sorsa, Katri Nissinen, Virpi Virjamo and Md. Nazmul Hasan for help in the lab, Anne Mehlhoop, Andreja Kovše and Umer Qureshi for soil sampling, Maria Greger for initial advice on soil sampling and phenolics, Gé van Steijn for comments on the analyses, and Jo Inge Breisjøberget and Kjell Anders Vikan for information about and digital maps based on the H40 system. We thank the Norwegian state-owned land and forest enterprise Statskog SF for permission to do the experiment on their property. MSG thanks the Stack Exchange Q&A web communities Stack Overflow and Cross Validated for invaluable statistical and analytical insights, Marieke Gonlag-Schrijvers for her patience and understanding, and Ole Arne Hagen / Linnea AS for facilitating a productive working environment during a pandemic.

## Author Contributions

**Conceptualization:** Marcel Schrijvers-Gonlag, Christina Skarpe.

**Data curation:** Marcel Schrijvers-Gonlag.

**Formal analysis:** Marcel Schrijvers-Gonlag, Riitta Julkunen-Tiitto.

**Funding acquisition:** Marcel Schrijvers-Gonlag, Christina Skarpe.

**Investigation:** Marcel Schrijvers-Gonlag, Christina Skarpe.

**Methodology:** Marcel Schrijvers-Gonlag, Christina Skarpe.

**Project administration:** Marcel Schrijvers-Gonlag.

**Resources:** Riitta Julkunen-Tiitto.

**Supervision:** Marcel Schrijvers-Gonlag, Christina Skarpe, Riitta Julkunen-Tiitto, Antonio B. S. Poléo.

**Validation:** Marcel Schrijvers-Gonlag.

**Visualization:** Marcel Schrijvers-Gonlag.

**Writing – original draft:** Marcel Schrijvers-Gonlag.

**Writing – review & editing:** Marcel Schrijvers-Gonlag, Christina Skarpe, Riitta Julkunen-Tiitto, Antonio B. S. Poléo.

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
