## [Decision Letter · Decision Letter 0]

27 Apr 2023

PONE-D-23-06554Phenolic concentrations and carbon/nitrogen ratio in annual shoots of bilberry (*Vaccinium myrtillus*) after simulated herbivoryPLOS ONE

Dear Dr. Schrijvers-Gonlag,

Thank you for submitting your manuscript to PLOS ONE. After careful consideration, we feel that it has merit but does not fully meet PLOS ONE’s publication criteria as it currently stands. Therefore, we invite you to submit a revised version of the manuscript that addresses the points raised during the review process.

The manuscript has been judged as too descriptive. The authors should put additional effort to reorganize it to gain scientific rigidity. Additional experimental setups and measurements might be needed to avoid speculations. Refocusing the manuscript to herbivory assessment only might be one of the major advices, as suggested by Reviewer #3.

The main editorial advice is to have the manuscript proofread by a senior researcher form the same field to diminish uninformative and redundant parts, by increasing its overall informativeness. Please be aware of the supplementary file provided by Reviewers #1 and #3.

We look forward to receiving your revised manuscript.

Kind regards,

Branislav T. Šiler, Ph.D.

Academic Editor

PLOS ONE

Journal Requirements:

Reviewers' comments:

Reviewer's Responses to Questions

**Comments to the Author**

1. Is the manuscript technically sound, and do the data support the conclusions?

Reviewer #1: Yes

Reviewer #2: Partly

Reviewer #3: Partly

2. Has the statistical analysis been performed appropriately and rigorously? 

Reviewer #1: Yes

Reviewer #2: Yes

Reviewer #3: No

3. Have the authors made all data underlying the findings in their manuscript fully available?

Reviewer #1: Yes

Reviewer #2: Yes

Reviewer #3: Yes

4. Is the manuscript presented in an intelligible fashion and written in standard English?

Reviewer #1: Yes

Reviewer #2: Yes

Reviewer #3: No

5. Review Comments to the Author

Reviewer #1: The manuscript entitled Phenolic concentrations and carbon/nitrogen ratio in annual shoots of bilberry (Vaccinium myrtillus) after simulated herbivory cover an interesting topic about the allocation of phenolic compounds during differents scenarios of herbivory. It is in general well written.

However there are some minor aspects that need to be taken to be ready to publish

Why no to use another statistic test such as GLM to test interactions between location and soil productivity

Don you think table 1 should be better in supplementary material?

I think is more informative to add the letters according to a post hoc test in table two. And why are you adding the column of all treatments together? It is useful??

Is it on the guidelines or why the figure legends are in the text?

I don’t think that is necessary to add SE and SD from you data every time you describe the results

I think first paragraph of discussion should be a summary of the results, why start with one particular compound?

When you make your comparisons with other systems, why do not you use related species instead of mountain birch and mopane? there are some interesting articles of other vaccinium species that can be more comparable to your study system

Reviewer #2: I reviewed the paper entitled ‘Phenolic concentrations and carbon/nitrogen ratio in annual shoots of bilberry (Vaccinium myrtillus) after simulated herbivory’. Authors identified several hypotheses related to phenolics in response to simulated herbivory. Principally, related to whether these defenses follow the C:nutrient balance hypothesis or Optimal Defense Theory. The actual experimental data was quite straightforward. Various phenolic compounds and C:N ratios were measured in bilberry shoots. That being said, in my view there is excessive text and lots of speculation that goes far beyond what was tested in this manuscript. The protein competition model is quite interesting, however not properly tested. I am not sure how you can test the protein competition model without measuring any protein…There are many reasons why phenolics would be reduced or unchanged. Did you see differences in protein quantity/quality under simulated herbivory? Further, the conclusion that this may be due to clonality is likewise interesting, but was not tested. Did you look at C:N ratios in clonal ramets connected to experimental ramets, for example, to see whether there may be a flux or allocation of resources to damaged plants. Additionally, why was the foliar content not measured? Especially considering you had a defoliation treatment, one might expect differences to be most apparent in leaves. Authors should justify why this was.

What was actually found, as I understand, is that, other than essentially complete removal of aboveground tissue, your treatments do not reduce specific constitutive phenolics, whereas under severe damage, shoots had lower phenolics, presumably because carbon is being used for regrowth of tissue growth and is limiting. Under ambient herbivory, carbon may not be limiting due to the fact that only minor additional growth is necessary to reach static conditions, and enough is C is thus available for phenolics biosynthesis. Did you have data for changes in biomass? Although the takeaway message is rather simple, I found the MS slightly challenging to read. And was distracted by a lot of possibilities and details that were not tested in the experiments. Perhaps if the paper was reframed and simplified to highlight what was tested, as opposed to what was not (all of these trade-off theories) it would be easier to follow. For example, your first prediction is not even possible to test, because you don’t detect any gallic acid-derived phenolics at all. As such, it doesn’t make much sense to include it in the MS. Your first prediction might instead be something like ‘determine whether bilberry has these compounds at all, and compare them with Phe-derived phenolics’. Then you might be able to contextualize this in the protein competition model. However, because they are not present, this is of course not possible.

I did not leave specific, detailed comments, as lines were not numbered. In subsequent drafts I would recommend adding line numbers.

Reviewer #3: After reading this study several times I can only conclude that to have a chance to be publishable, the manuscript needs to be completely rewritten to address two major problems: 1) a weak initial study design that prevents conclusions to be made with confidence followed by a modification that does little to improve the scientific integrity, and 2) a writing style that is difficult to follow due to a lack of flow and clarity arising from awkwardly constructed sentences and arguments, excessive repetition and inclusion of unnecessary text and information.

With respect strengthening the design of the study, I recommend removing the issue of high- and low productivity sites. The definition of productivity appears to be defined and supported data-wise with one variable, total nitrogen. This information comes from one composite sample per site (n=1). To merge your data and compare this to the one location which “behaved differently” also weakens your study in my opinion. I suggest you remove the idea of site productivity as a factor and rewrite the manuscript based on your “herbivory treatments”; something for which you have much better resolution and data from which to draw more confident conclusions. Present the plant chemistry data (phenols, tannins, N, C, and C/N) together in one Table or Figure, so that the reader can better visualize possible relationships among these variables (for example, see Nosko et al 2020, https://doi.org/10.1016/j.foreco.2019.117839). In your discussion, you can then focus on browsing treatments as they affect the relationships among plant chemistry variables. In the absence of such relationships, some inspiration might be obtained from Warbrick et al 2020 https://doi.org/10.1007/s11258-020-01027-y and Nosko and Embury 2018 https://doi.org/10.1007/s11258-018-0821-7.

The manuscript is very difficult to read and needs to be completely rewritten and shortened to improve readability, flow and clarity. Every element, including captions, needs to be edited and reorganized. Reading through comparisons among sites/treatments is quite cumbersome (see examples on the manuscript). Flow and clarity could be greatly improved by adopting a code or acronym for your site and treatment names (see suggestions in edited manuscript). Flow and length will also be improved by removing site productivity as a factor and presenting plant chemistry data together. I did not provide editorial suggestions for the entire manuscript (see attached), but did so in several sections. Hopefully this will give you an idea of my concerns.

6. PLOS authors have the option to publish the peer review history of their article (what does this mean?). If published, this will include your full peer review and any attached files.

Reviewer #1: **Yes: **JHC

Reviewer #2: No

Reviewer #3: No

---

## [Author Response · Author response to Decision Letter 0]

6 Aug 2023

All answers to all comments are within the document that I ioloaded, titled: Response to Reviewers - MSG20230804.docx

As you ask for this here as well, I'll copy my text below:

PONE-D-23-06554

Phenolic concentrations and carbon/nitrogen ratio in annual shoots of bilberry (Vaccinium myrtillus) after simulated herbivory

PLOS ONE

Response to Reviewers

In this rebuttal letter, I respond to points raised in the review process. First, I repeat the issue/point raised. My response follows after 'Response MSG:', font colour blue, see below.

With regards,

Marcel Schrijvers-Gonlag

Campus Evenstad, Faculty of Applied Ecology, Agricultural Sciences and Biotechnology

Inland Norway University of Applied Sciences, Koppang, Norway

marcel.schrijversgonlag@inn.no

Journal Requirements:

Response MSG: I have studied the style templates in detail and have used these style requirements in the manuscript.

Response MSG: You ask for correct grant numbers. We used funding from two grants, as stated in the Financial Disclosure. The grant number for the first one is given, but the second grant (from the Extensus Foundation) does not have a grant number. Therefore, no number is provided.

Response MSG: no comment.

Response MSG: I have included the ethics statement in the 'Methods' section.

Response MSG: I have removed Figure 1 completely. 

Comments to the Author

1. Is the manuscript technically sound, and do the data support the conclusions?

Reviewer #1: Yes

Reviewer #2: Partly

Reviewer #3: Partly

Response MSG: The manuscript has been changed according to many reviewer recommendations (under '5. Review Comments to the Author').

2. Has the statistical analysis been performed appropriately and rigorously? 

Reviewer #1: Yes

Reviewer #2: Yes

Reviewer #3: No

Response MSG: The manuscript has been changed according to many reviewer recommendations (under '5. Review Comments to the Author').

3. Have the authors made all data underlying the findings in their manuscript fully available?

Reviewer #1: Yes

Reviewer #2: Yes

Reviewer #3: Yes

Response MSG: no comment.

4. Is the manuscript presented in an intelligible fashion and written in standard English?

Reviewer #1: Yes

Reviewer #2: Yes

Reviewer #3: No

Response MSG: The manuscript has been changed according to many reviewer recommendations (under '5. Review Comments to the Author').

5. Review Comments to the Author

Reviewer #1: The manuscript entitled Phenolic concentrations and carbon/nitrogen ratio in annual shoots of bilberry (Vaccinium myrtillus) after simulated herbivory cover an interesting topic about the allocation of phenolic compounds during differents scenarios of herbivory. It is in general well written.

However there are some minor aspects that need to be taken to be ready to publish

Why no to use another statistic test such as GLM to test interactions between location and soil productivity

Response MSG: The variable Soil productivity has been removed from the manuscript (as suggested by reviewer #3).

Don you think table 1 should be better in supplementary material?

Response MSG: Table 1 is about soil nitrogen. The variable Soil productivity has been removed from the manuscript (as directly suggested by reviewer #3 and indirectly by reviewer #1). This means that also Table 1 has been removed from the manuscript.

I think is more informative to add the letters according to a post hoc test in table two. And why are you adding the column of all treatments together? It is useful??

Response MSG: There is some overlap between Table 2 and Figure 2. I have removed Table 2 and kept the data for all threatments together Table 1 (this is useful information as it indicates the range of all data). The post hoc test letters are in Figure 2 (which is now Figure 1 in the revised manuscript).

Is it on the guidelines or why the figure legends are in the text?

Response MSG: Correct, it says in the PLoS ONE guidelines that the figure legends should be in the text (https://journals.plos.org/plosone/s/figures#loc-how-to-submit-figures-and-captions).

I don’t think that is necessary to add SE and SD from you data every time you describe the results

Response MSG: When presenting mean values, it can be very useful, although not necessary, to give information about either the spread of the data (by presenting the SD) or information about how far the mean is likely to be from the population mean (the 'real value') (by presenting the SE). For each mean value presented I have chosen to include the most appropriate parameter: the SD, the SE, or both.

I think first paragraph of discussion should be a summary of the results, why start with one particular compound?

Response MSG: I have changed the discussion, now it starts with a summary of the results.

When you make your comparisons with other systems, why do not you use related species instead of mountain birch and mopane? there are some interesting articles of other vaccinium species that can be more comparable to your study system

Response MSG: Despite an extensive search I was not able to locate papers with other Vaccinium species that addresses the particular topic that is discussed in this paragraph, therefore I used papers with Mountain birch and Mopane who do address the particular topic instead. To satisfy some reviewers' requests, I have removed this particular text.

The supplementary file provided by Reviewer #1 contained many textual suggestions and some questions. I have answered the questions in the text and changed the manuscript accordingly, when appropriate. Some changes were not necessary and more a matter of taste, in these cases I often did not change my own words. As the text concerning the variable Soil productivity has been removed in the new version of the manuscript, many comments about this part can be neglected.

Reviewer #2: I reviewed the paper entitled ‘Phenolic concentrations and carbon/nitrogen ratio in annual shoots of bilberry (Vaccinium myrtillus) after simulated herbivory’. Authors identified several hypotheses related to phenolics in response to simulated herbivory. Principally, related to whether these defenses follow the C:nutrient balance hypothesis or Optimal Defense Theory. The actual experimental data was quite straightforward. Various phenolic compounds and C:N ratios were measured in bilberry shoots. That being said, in my view there is excessive text and lots of speculation that goes far beyond what was tested in this manuscript. The protein competition model is quite interesting, however not properly tested. I am not sure how you can test the protein competition model without measuring any protein…There are many reasons why phenolics would be reduced or unchanged. Did you see differences in protein quantity/quality under simulated herbivory? 

Response MSG: I have removed the Protein Competition Model and all text concerning proteins from the Introduction chapter and mentioned the PCM briefly in the Discussion chapter.

Further, the conclusion that this may be due to clonality is likewise interesting, but was not tested. Did you look at C:N ratios in clonal ramets connected to experimental ramets, for example, to see whether there may be a flux or allocation of resources to damaged plants. 

Response MSG: Clonality is only mentioned in the Discussion section, and is not mentioned in the Conclusion section. We did not look at C:N ratios in clonal ramets as this was not the goal of our study.

Additionally, why was the foliar content not measured? Especially considering you had a defoliation treatment, one might expect differences to be most apparent in leaves. Authors should justify why this was.

Response MSG: We did not measure foliar content as our study focused on the influence of simulated herbivory (on leaves, but also more severe herbivory on shoots) on phenolic content and C:N ratio in annual shoots. One of the reasons that we focused on annual shoots instead of leaves was that many other studies have focused on leaves but studies on annual shoots were scarce.

What was actually found, as I understand, is that, other than essentially complete removal of aboveground tissue, your treatments do not reduce specific constitutive phenolics, whereas under severe damage, shoots had lower phenolics, presumably because carbon is being used for regrowth of tissue growth and is limiting. Under ambient herbivory, carbon may not be limiting due to the fact that only minor additional growth is necessary to reach static conditions, and enough is C is thus available for phenolics biosynthesis. Did you have data for changes in biomass?

Response MSG: We did measure several morphological properties of all bilberry ramets, including biomass of annual shoots, but these data will be used in another manuscript. In the reviewed manuscript we focus on chemical responses following herbivory, in a future manuscript we will focus on morphological responses following herbivory.

Although the takeaway message is rather simple, I found the MS slightly challenging to read. And was distracted by a lot of possibilities and details that were not tested in the experiments. Perhaps if the paper was reframed and simplified to highlight what was tested, as opposed to what was not (all of these trade-off theories) it would be easier to follow. For example, your first prediction is not even possible to test, because you don’t detect any gallic acid-derived phenolics at all. As such, it doesn’t make much sense to include it in the MS. Your first prediction might instead be something like ‘determine whether bilberry has these compounds at all, and compare them with Phe-derived phenolics’. Then you might be able to contextualize this in the protein competition model. However, because they are not present, this is of course not possible.

Response MSG: I have removed the first prediction and the Protein Competition Model from the Introduction chapter and mentioned the PCM briefly in the Discussion chapter. Furthermore, I have adjusted the text at many places in order to make it easier to read.

I did not leave specific, detailed comments, as lines were not numbered. In subsequent drafts I would recommend adding line numbers.

Response MSG: Line numbers were present in the manuscript but unfortunately disappeared just for submission (possibly due to a Word update). I found out about this just after submission and contacted the PLoS ONE editor instandly. I got the following reply: '... I believe the reviewers will find their way to address targeted comments. If they encounter problems regarding line numbering, I will get back to you.'

In the revised manuscript I have added line numbers.

Reviewer #3: After reading this study several times I can only conclude that to have a chance to be publishable, the manuscript needs to be completely rewritten to address two major problems: 1) a weak initial study design that prevents conclusions to be made with confidence followed by a modification that does little to improve the scientific integrity, and 2) a writing style that is difficult to follow due to a lack of flow and clarity arising from awkwardly constructed sentences and arguments, excessive repetition and inclusion of unnecessary text and information.

With respect strengthening the design of the study, I recommend removing the issue of high- and low productivity sites. The definition of productivity appears to be defined and supported data-wise with one variable, total nitrogen. This information comes from one composite sample per site (n=1). To merge your data and compare this to the one location which “behaved differently” also weakens your study in my opinion. I suggest you remove the idea of site productivity as a factor and rewrite the manuscript based on your “herbivory treatments”; something for which you have much better resolution and data from which to draw more confident conclusions.

Response MSG: The variable Soil productivity has been removed from the manuscript. Also, the two groups that were analysed separately in the reviewed manuscript, have been merged into one dataset, and all analyses have been adjusted accordingly, as suggested by reviewer #3. Furthermore, I have adjusted the text at many places in order to make it easier to read.

Present the plant chemistry data (phenols, tannins, N, C, and C/N) together in one Table or Figure, so that the reader can better visualize possible relationships among these variables (for example, see Nosko et al 2020, https://doi.org/10.1016/j.foreco.2019.117839).

Response MSG: I have taken over this suggestion and many chemistry data (tannins, all phenolics, C, N, C/N) are now in one table (Table 1).

In your discussion, you can then focus on browsing treatments as they affect the relationships among plant chemistry variables. In the absence of such relationships, some inspiration might be obtained from Warbrick et al 2020 https://doi.org/10.1007/s11258-020-01027-y and Nosko and Embury 2018 https://doi.org/10.1007/s11258-018-0821-7.

Response MSG: I have changed the discussion according to several recommendations from the reviewers.

The manuscript is very difficult to read and needs to be completely rewritten and shortened to improve readability, flow and clarity. Every element, including captions, needs to be edited and reorganized.

Response MSG: I have adjusted the text at many places in order to make it easier to read. For this, the many adjustments made by Reviewer #1 and Reviewer #3 have been very useful.

Reading through comparisons among sites/treatments is quite cumbersome (see examples on the manuscript). Flow and clarity could be greatly improved by adopting a code or acronym for your site and treatment names (see suggestions in edited manuscript). 

Response MSG: The suggested codes/acronyms in the edited manuscript have been used in the revised manuscript.

Flow and length will also be improved by removing site productivity as a factor and presenting plant chemistry data together. 

Response MSG: The variable Soil productivity has been removed from the manuscript and most chemistry data are now in one table (Table 1).

I did not provide editorial suggestions for the entire manuscript (see attached), but did so in several sections. Hopefully this will give you an idea of my concerns.

Response MSG: All suggestions for improvement have been considered. As with Reviewer #1, I have changed the manuscript accordingly, when appropriate.

6. PLOS authors have the option to publish the peer review history of their article (what does this mean?). If published, this will include your full peer review and any attached files.

Do you want your identity to be public for this peer review? For information about this choice, including consent withdrawal, please see our Privacy Policy.

Reviewer #1: Yes: JHC

Reviewer #2: No

Reviewer #3: No

---

## [Decision Letter · Decision Letter 1]

12 Dec 2023

PONE-D-23-06554R1Phenolic concentrations and carbon/nitrogen ratio in annual shoots of bilberry (*Vaccinium myrtillus*) after simulated herbivory.PLOS ONE

Dear Dr. Schrijvers-Gonlag,

Thank you for submitting your manuscript to PLOS ONE. After careful consideration, we feel that it has merit but does not fully meet PLOS ONE’s publication criteria as it currently stands. Therefore, we invite you to submit a revised version of the manuscript that addresses the points raised during the review process.

Reviewer #2 raised several very important concerns regarding the tradeoff existing in changing herbivory regimes between defense and metabolic processes. Please take into a careful consideration their points, which can aid in further improvement of the manuscript quality. Furthermore, a visual presentation of the study setup would be much helpful.==============================

We look forward to receiving your revised manuscript.

Kind regards,

Branislav T. Šiler, Ph.D.

Academic Editor

PLOS ONE

Reviewers' comments:

Reviewer's Responses to Questions

**Comments to the Author**

1. If the authors have adequately addressed your comments raised in a previous round of review and you feel that this manuscript is now acceptable for publication, you may indicate that here to bypass the “Comments to the Author” section, enter your conflict of interest statement in the “Confidential to Editor” section, and submit your "Accept" recommendation.

Reviewer #2: (No Response)

2. Is the manuscript technically sound, and do the data support the conclusions?

Reviewer #2: Partly

3. Has the statistical analysis been performed appropriately and rigorously? 

Reviewer #2: Yes

4. Have the authors made all data underlying the findings in their manuscript fully available?

Reviewer #2: Yes

5. Is the manuscript presented in an intelligible fashion and written in standard English?

Reviewer #2: Yes

6. Review Comments to the Author

Reviewer #2: This is my second time reviewing this paper, and while I appreciate that the authors have made substantial revisions to their MS, which have certainly increased the overall quality and clarity of the paper, I still have some concerns regarding the data presentation and the claims made. Namely, there is substantial mention of growth, for example, lines 60, 369 and 459. While I understand authors may want to use biomass data (which they have admittedly collected) for a second MS, I find that it is an important piece of data that would make their claims and discussions more robust and compelling. Especially if the aim is to, at least in part, understand the tradeoffs between defense and other metabolic processes across variable herbivory regimes. For example, you can not say for certain that plants are prioritizing growth more or less between different defoliation treatments because you have no metric of growth (total biomass, compensatory growth). To me this is extremely relevant. One scenario I could think of is that the loss of biomass from mild herbivory treatments is effectively inconsequential for plant fitness and thus responses are not obvious. When biomass loss becomes too great, plants must begin to divert resources from other pools to compensate for biomass losses. It would be interesting to see the differences in growth after herbivory treatments, which would determine if severely damaged plants actually put on more biomass then more moderately damaged plants, or if simply the amount of resources are so low that these severely damaged plants not only produce less phenolics, but also less growth and thus metabolize less, generally. Also, while the C and N concentrations may remain stable across treatments, the total pool of each that plants have access to is not consistent; in this case it is likely a function of biomass. So the amount of carbon severely damaged plants have access to should be less then mildly-damaged plants if there is a biomass reduction.

Additionally, I find the entire conclusion section quite difficult to follow. What exactly is the takeaway message?

Finally, perhaps a diagram of the plant treatments would be helpful. It is difficult to determine exactly how your treatments were conducted. As such, I can't help but wonder if some of your results could be an artifact of some concentration/dilution effect by sampling different numbers of stems/ramets between treatments, that is more severe with increasing herbivory. A clear outline (visual) could help to assuage these concerns.

7. PLOS authors have the option to publish the peer review history of their article (what does this mean?). If published, this will include your full peer review and any attached files.

Reviewer #2: No

---

## [Author Response · Author response to Decision Letter 1]

19 Jan 2024

6. Review Comments to the Author

Reviewer #2: This is my second time reviewing this paper, and while I appreciate that the authors have made substantial revisions to their MS, which have certainly increased the overall quality and clarity of the paper, I still have some concerns regarding the data presentation and the claims made. Namely, there is substantial mention of growth, for example, lines 60, 369 and 459. While I understand authors may want to use biomass data (which they have admittedly collected) for a second MS, I find that it is an important piece of data that would make their claims and discussions more robust and compelling. Especially if the aim is to, at least in part, understand the tradeoffs between defense and other metabolic processes across variable herbivory regimes. For example, you can not say for certain that plants are prioritizing growth more or less between different defoliation treatments because you have no metric of growth (total biomass, compensatory growth). To me this is extremely relevant. One scenario I could think of is that the loss of biomass from mild herbivory treatments is effectively inconsequential for plant fitness and thus responses are not obvious. When biomass loss becomes too great, plants must begin to divert resources from other pools to compensate for biomass losses. It would be interesting to see the differences in growth after herbivory treatments, which would determine if severely damaged plants actually put on more biomass then more moderately damaged plants, or if simply the amount of resources are so low that these severely damaged plants not only produce less phenolics, but also less growth and thus metabolize less, generally. Also, while the C and N concentrations may remain stable across treatments, the total pool of each that plants have access to is not consistent; in this case it is likely a function of biomass. So the amount of carbon severely damaged plants have access to should be less then mildly-damaged plants if there is a biomass reduction.

Response MSG: we have been in contact with PLoS ONE by email (12 January 2024), as we cannot meet this point. The reply I got from PLoS ONE is below (received on 15 January 2024):

"I agree that the suggested morphometric evaluation would be an

exhausting job and further delay the manuscript publication. Although I

strongly support the Reviewer #2's demand for additional experimental

assessment, which would remarkably contribute the study

comprehensiveness, I find the manuscript in this form contains a fair

dose of novel scientific information that candidate it for acceptance

for publication if other reviewer's requirements are to be met. However,

I urge the authors to properly discuss in the text the limitations of

the present findings in light of Reviewer #2's comments."

Best regards,

Branislav

Response MSG: Therefore, I have added a paragraph in the Discussion section, called ‘Defense and other metabolic processes’, to discuss this issue.

Additionally, I find the entire conclusion section quite difficult to follow. What exactly is the takeaway message?

Response MSG: The conclusion section has been changed, to clarify our intention.

Finally, perhaps a diagram of the plant treatments would be helpful. It is difficult to determine exactly how your treatments were conducted. As such, I can't help but wonder if some of your results could be an artifact of some concentration/dilution effect by sampling different numbers of stems/ramets between treatments, that is more severe with increasing herbivory. A clear outline (visual) could help to assuage these concerns.

Response MSG: I have added a new Figure 1, which illustrates the study design.

---

## [Editor Report · Decision Letter 2]

22 Jan 2024

Phenolic concentrations and carbon/nitrogen ratio in annual shoots of bilberry (*Vaccinium myrtillus*) after simulated herbivory.

PONE-D-23-06554R2

Dear Dr. Schrijvers-Gonlag,

We’re pleased to inform you that your manuscript has been judged scientifically suitable for publication and will be formally accepted for publication once it meets all outstanding technical requirements.

Kind regards,

Branislav T. Šiler, Ph.D.

Academic Editor

PLOS ONE
---

## [Editor Report · Acceptance letter]

23 Feb 2024

PONE-D-23-06554R2 

PLOS ONE

Dear Dr. Schrijvers-Gonlag, 

I'm pleased to inform you that your manuscript has been deemed suitable for publication in PLOS ONE. Congratulations! Your manuscript is now being handed over to our production team.

Kind regards, 

on behalf of

Dr. Branislav T. Šiler 

Academic Editor

PLOS ONE